# Overexpression of Two MADS-Box Genes from *Lagerstroemia speciosa* Causes Early Flowering and Affects Floral Organ Development in *Arabidopsis*

**Lichen Yang** [1,2,3,4,5,6,†], **Zhuojiao Li** [1,2,3,4,5,6,†], **Tangchun Zheng** [1,2,3,4,5,6,*], **Jia Wang** [1,2,3,4,5,6], **Tangren Cheng** [1,2,3,4,5,6] **and Qixiang Zhang** [1,2,3,4,5,6,*]

1   Beijing Key Laboratory of Ornamental Plants Germplasm Innovation and Molecular Breeding, Beijing Forestry University, Beijing 100083, China
2   National Engineering Research Center for Floriculture, Beijing Forestry University, Beijing 100083, China
3   Beijing Laboratory of Urban and Rural Ecological Environment, Beijing Forestry University, Beijing 100083, China
4   Engineering Research Center of Landscape Environment of Ministry of Education, Beijing Forestry University, Beijing 100083, China
5   Key Laboratory of Genetics and Breeding in Forest Trees and Ornamental Plants of Ministry of Education, Beijing Forestry University, Beijing 100083, China
6   School of Landscape Architecture, Beijing Forestry University, Beijing 100083, China
*   Correspondence: zhengtangchun@bjfu.edu.cn (T.Z.); zqx@bjfu.edu.cn (Q.Z.); Tel.: +86-10-62336321 (T.Z.); +86-10-62338005 (Q.Z.)
†   These authors contributed equally to this work.

**Abstract:** *Lagerstroemia speciosa* is an important ornamental plant, and there is only one double-petaled variety, 'Yunshang', produced by natural mutation, in the whole genus of *Lagerstroemia*. The members of the MADS-box family play important roles in floral organ development. However, little is known about the biological function of the MADS-box gene in *L. speciosa*. In our study, two MADS-box genes (*LsAG2* and *LsDEF1*) were isolated from *L. speciosa*, and their expression levels in different tissues and developmental stages were analyzed by RT-qPCR. Subcellular localization showed that LsAG2 and LsDEF1 are localized in the nucleus. The overexpression of *LsAG2* and *LsDEF1* in *Arabidopsis thaliana* caused transgenic plants to exhibit different phenotypes, such as floral organ aberrations, early flowering, and dwarf plants, and resulted in the up-regulation of endogenous genes related to flowering (i.e., *AP1*, *PI*, *FLC*, *FUL*, *LFY*, and *FT*). Our results provide a theoretical basis for the flowering time, flower development, and genetic improvement of double-petaled flowers in the genus of *Lagerstroemia*.

**Keywords:** *Lagerstroemia speciosa*; MADS-box gene; *LsAG2*; *LsDEF1*; flowering time; flower development

## 1. Introduction

A flower is the reproductive organ of angiosperms. Studying the flower developmental process of ornamental plants is of great significance for measures aiming to adjust the flowering period and flower shape, and to improve the ornamental and economic value of the plants. Floral meristem determination is a complex developmental event that is regulated by a variety of genetic factors [1]. The flower of *A. thaliana* is typically representative of angiosperm flowers. The flower development process of *A. thaliana* was initially summarized as the "ABC model" [2] and then gradually expanded to the "ABCDE model" [3]. The "ABC model" divides the genes involved in flower organ differentiation into three categories: A, B, and C, which, respectively, regulate the development of the calyx, petal, stamen, and pistil, either separately or coordinately. In addition, *Arabidopsis* mutations in class A and class C genes lead these genes to exhibit ectopic expressions of each other, thereby altering the flower morphology and suggesting that the class A and C

genes are antagonistic to each other [4,5]. In the study of *Petunia hybrida* flower development, the newly proposed class D genes were proven to be involved in the regulation of *P. hybrida* ovule formation [6], and the *A. thaliana* homologous genes of the class D gene in *P. hybrida* have also had their involvement in the regulation of the development and formation of *A. thaliana* ovules [7] confirmed. Moreover, researchers have found that if the functions of *SEP1* (*SEPALLATA1*), *SEP2*, and *SEP3* in *A. thaliana* are simultaneously inhibited, all the floral organs of *Arabidopsis* will be completely changed to sepals [8]. There is speculation that these genes are independent of ABC genes and have regulatory effects on the expression level of ABC genes. These genes were named as E-type genes, which are considered to be involved in floral organ regulation during flower development and floral transition [9].

The different flower forms of angiosperms, especially the double petal trait, can greatly improve the ornamental value of plants. There are many origins of existing double-petaled ornamental plants, and the existing studies on a variety of plants that naturally form double flowers and single flowers have shown that the formation of double flowers is mostly related to changes in the plant MADS-box gene sequence or expression. For example, the *Arabidopsis* mutant resulting from *AG* (*AGAMOUS*) gene deletion shows a double-petal phenotype [10], and the formation of the double-petal variety 'Double White' in *Thalictrum thalictroides* is due to the deletion of the K domain of the *ThtAG1* gene transcription protein, which affects interactions with the E-type protein ThtSEP3 [11].

Studies on the MADS-box family genes have shown that members of this family are also widely involved in the transformation of plants into flowers and the formation of the flower organ morphology. For example, the overexpression of two *SUPPRESSOR OF OVEREXPRESSION OF CONSTANS 1* (*SOC1*) homologs in *Pyrus bretschneideri* leads to early flowering in *Arabidopsis* [12]. Most of the other genes affect the development of plant flowers by participating in the regulation of MADS-box genes. For example, the *LEAFY* (*LFY*) gene in *A. thaliana* interacts with the *APETALA1* (*AP1*) gene to promote the transformation of inflorescences into floral meristems and participates in the determination of floral organ meristems through the regulation of *APETALA3* (*AP3*) and *PISTILLATA* (*PI*) [13]. The *WUSCHEL* (*WUS*) gene regulates floral organ determination by inducing the expression of the *AG* gene, while *AG* counteracts *WUS* and inhibits its role in floral meristem differentiation [14]. Among the ABCE-type genes, there are some *APETALA2* (*AP2*) family genes, while the other genes are MIKC-type MADS-box family genes [15,16]. With the advancement of whole-genome sequencing technology, the MADS-box family genes of many ornamental plants have been identified and analyzed, including carnations (*Dianthus caryophyllus*) [17], wild chrysanthemum (*Chrysanthemum nankingense*) [18], mei (*Prunus mume*) [19], and so on.

Crape myrtle is an ornamental woody plant used worldwide. It blooms in midsummer, and the flowering period is as long as three months. *Lagerstroemia speciosa* is a special species of the *Lagerstroemia* genus, among which the flowers of *L. speciosa* are very large (5–7 cm in diameter) and colorful, with high garden application and ornamental value. There are approximately 55 species in the whole *Lagerstroemia* genus and approximately 200 cultivated varieties of crape myrtle, but there is only one double-petaled variety, 'Yunshang', produced by a natural bud mutation of *L. speciosa*, with defined double petals resulting from stamen petalization. Differential transcriptome sequencing of single-petaled *L. speciosa* and double-petaled *L. speciosa* at different stages of flower organ development was carried out, and four flower-development-related genes were identified [20]. However, our knowledge of the biological function of the flower-development-related genes in *Lagerstroemia* is still limited. In our study, we cloned two MADS-box family genes related to the regulation of flower organ development in *L. speciosa*. The temporal and spatial expression patterns of the two genes were analyzed, and the biological functions were further verified in *Arabidopsis*, which provided a theoretical basis for the flower time, flower development, and genetic improvement of double-petal flowers in *Lagerstroemia*.

## 2. Materials and Methods

### 2.1. Plant Material

Plant materials were obtained from the resource garden of Guangxi Academy of Forestry (Nanning City, Guangxi Province). The samples were quick-frozen in liquid nitrogen and then stored at $-80\ ^\circ$C in a freezer. The double-petaled *L. speciosa* (DPL) is derived from the single-petaled *L. speciosa* (SPL). Flower buds of different developmental stages (S1: buds 4–6 mm in diameter, S2: buds 6–10 mm in diameter, S3: buds 10–14 mm in diameter, and S4: full bloom but not powered) and different floral organ tissues (calyx, petals, stamens/petalized stamens, and pistils) at S4 stage of the SPL and DPL were collected, as in our previous study [20]. In total, 16 samples were used to analyze the gene expression patterns of MADS-box genes related to flower development in *L. speciosa*.

*Arabidopsis thaliana* (Col-0) plants were grown in an artificial climate chamber with an average temperature of $22 \pm 2\ ^\circ$C, with 16 h of light and 8 h of darkness, $100\ \mu\text{mol}\cdot\text{m}^{-2}\cdot\text{s}^{-1}$ light intensity, and 65–75% relative humidity.

### 2.2. Sequence Analysis and Phylogenetic Tree Construction

Physicochemical properties such as the molecular weight (MW) and theoretical isoelectric point (*pI*) were predicted in Prot Param (https://web.expasy.org/protparam/, accessed on 7 May 2020). Protein hydrophobicity was predicted with an online tool in Prot Scale (https://web.expasy.org/protscale/, accessed on 7 May 2020). The SOPMA online software (https://npsa-prabi.ibcp.fr/cgi-bin/npsa_automat.pl?page=npsa_sopma.html, accessed on 7 May 2020) was used to predict the secondary structures of the proteins. The conserved protein domains of LsAG2 and LsDEF1 were predicted using an online tool in NCBI (https://www.ncbi.nlm.nih.gov/cdd, accessed on 7 May 2020). Phosphorylation sites were predicted in NetPhos (https://services.healthtech.dtu.dk/service.php?NetPhos-3.1, accessed on 10 May 2020). Plant-mPLoc in Cell-PLoc 2.0 (https://services.healthtech.dtu.dk/service.php?SignalP-5.0, accessed on 10 May 2020) was used for subcellular localization analysis. TMHMM Server 2.0 was employed to predict transmembrane helices in the proteins (https://services.healthtech.dtu.dk/service.php?TMHMM-2.0, accessed on 10 May 2020). The results of the gene structure were visualized in GSDS 2.0 (http://gsds.gao-lab.org/, accessed on 10 May 2020). We used the NCBI Blastx website (https://blast.ncbi.nlm.nih.gov/Blast.cgi, accessed on 10 May 2020) to align the gene-encoded protein sequences with other plant protein sequences in the database and the NJ method to construct a phylogenetic tree between the two gene-encoded proteins and the aligned homologous proteins.

### 2.3. RNA Extraction and RT-qPCR Analysis

A total of 16 samples representing 4 different developmental stages and 4 different floral organ tissues of the DPL and SPL were collected. Total RNA was extracted from the 16 samples using an RNAprep Pure Plant Kit (TianGen, Beijing, China). First-strand cDNAs were synthesized using a PrimeScript$^{\text{TM}}$ RT Reagent Kit (TaKaRa, Dalian, China), according to the manufacturer's instructions.

RT-qPCR was performed using TB Green$^{\circledR}$ Premix Ex Taq™ II (TaKaRa) and a CFX Connect Real-Time System (Bio-Rad, Hercules, CA, USA). The primers for RT-qPCR (Table S1) were designed using Integrated DNA Technologies tools (https://sg.idtdna.com/scitools/Applications/RealTimePCR/, accessed on 15 July 2020) and synthesized by Sangon Biotech (Shanghai, China) Co., Ltd. RT-qPCR was performed in 20 μL volumes containing 10 μL of TB Green Premix, 0.8 μL of forward primer (10 μM), 0.8 μL of reverse primer (10 μM), 2 μL of cDNA template (50 ng/μL), and 6.4 μL ddH$_2$O. Each test included three technical repetitions and three biological replicates. The *Elongation factor-1-alpha* (*EF-1α*) gene of *L. speciosa* was employed as a reference gene [21]. For the same type of tissue or developmental stage, the sample with the lowest expression was selected as the control, and the gene expression levels were calculated using the $2^{-\Delta\Delta\text{Ct}}$ method [22].

The total RNA was isolated from whole flowers of transgenic and wild-type plants using a MiniBEST Plant RNA Extraction Kit (TaKaRa), and we detected the expression levels of the *LsAG2* and *LsDEF1* genes in the transgenic plants and the endogenous flower-development-related genes in *A. thaliana*. Specific primers (Table S1) were designed for *AP1*, *FRUITFULL* (*FUL*), *LFY*, *FLOWERING LOCUS T* (*FT*), *FLOWERING LOCUS C* (*FLC*), and *PI*. Then, RT-qPCR analysis was performed, with the *Actin* gene employed as a reference gene [23].

### 2.4. Cloning of the Two MADS-Box Genes

The specific primers (Table S1) of *LsAG2* and *LsDEF1* were designed using Primer premier 5.0 software, according to the transcriptome data. Then, the sequences of *LsAG2* and *LsDEF1* were amplified from the cDNA of the SPL and DPL flower buds with specific primers (Table S1). The purified PCR products were constructed into a pCloneEZ Blunt Cloning vector (Taihe Biotechnology, Beijing, China), and the final LsAG2 and LsDEF1 sequences were further verified by sequencing (Sangon Biotech Co., Ltd., Shanghai, China).

### 2.5. Subcellular Localization Analysis

To analyze the subcellular localization of *LsAG2* and *LsDEF1*, the CDSs (without a stop codon) of *LsAG2* and *LsDEF1* were cloned into the vector pSuper1300-GFP to generate the *LsAG2-GFP* and *LsDEF1-GFP* fusion genes driven by *CaMV35S*, respectively. The recombinant plasmids pSuper1300-*LsAG2-GFP* and pSuper1300-*LsDEF1-GFP* were transformed into *Agrobacterium tumefaciens* competent cells, GV3101, and the recombinant *Agrobacterium* and empty vector control were injected into healthy *Nicotiana benthamiana* leaves, respectively. The infected tobacco plants were cultured in the dark for 24 h and then placed in an incubator for normal long-day culture for 2–3 d. Finally, the green fluorescent signals were observed using a TCS SP8 confocal microscope (Leica, Wetzlar, Germany). 4′,6-diamidino-2-phenylindole (DAPI) was used to dye the nuclear DNA as a positive control for nuclear localization.

### 2.6. Vector Construction and Plant Transformation

The plant expression vector was constructed using the method of seamless cloning, and the PCR primers (Table S1) of the target fragments were synthesized according to the instructions for the CV12-Seamless Assembly and Cloning kit (Aidlab, Beijing, China). With pCAMBIA1304 linearized by double digestion using restriction enzymes, *Nco* I and *Bst* E II, the PCR fragments of *LsAG2* and *LsDEF1* were inserted into the linearized vector to form the recombinant vectors (pCAMBIA1304-*LsAG2*, pCAMBIA1304-*LsDEF1*). Finally, the recombinant vectors were transferred into GV3101 using the freeze–thaw transformation method and transformed into *A. thaliana* via the floral dip method.

Transgenic *Arabidopsis* seeds were selected and placed on 1/2 MS medium containing 50 mg/L Kanamycin. After these positive seedlings developed four true leaves, they were then transferred into pots containing a mixture of turf peat, vermiculite, and pearlite (3:1:1 *v/v*) in an artificial climate chamber.

### 2.7. Phenotypic Observation

The number of days that a single flower lasted from the beginning to the end of the experiment was regarded as the flowering time. The phenotypes of transgenic lines were observed every 2 d from the opening of the first flower, such as flower and leaf deformity variation, etc., and we recorded indicators such as the flower volume and flowering period. The flowering period refers to the time from the blooming of the first flower to the fading of the last flower.

*2.8. Statistical Analyses*

Statistical analyses were performed using the SPSS 19.0 (SPSS, Chicago, IL, USA). ANOVA was used for multiple-group comparisons, and statistical significance ($p < 0.05$) was determined by Student's *t*-test.

## 3. Results

*3.1. Cloning of LsAG2 and LsDEF1 and Bioinformatics Analysis*

On the basis of the differentially expressed genes (DEGs) derived from previous transcriptome data, two genes related to flower development in the MADS-box family of *L. speciosa* were cloned. Comparing the CDS and DNA sequences of these two genes cloned from the SPL and DPL, it was found that there was no difference in the amino acid sequences, indicating that the formation of stamen petalization in *L. speciosa* was not caused by the gene structure differences between *LsAG2* and *LsDEF1*. Prediction of the upstream promoter sequences of these genes showed that the basic cis-elements, such as TATA-box and CAAT-box, were found in *LsAG2* and *LsDEF1*, and there are multiple cis-acting elements involved in the hormone response and light response (Table S2). In terms of gene structure, *LsAG2* and *LsDEF1* contain 6 and 0 intron regions, respectively, and encode proteins of 237 and 252 amino acids in length, with relative molecular weights of 27.2 kDa and 29.07 kDa, respectively. These two proteins are hydrophilic, basic, and unstable proteins. They do not contain signal peptide sites or transmembrane domains and are not secreted proteins. LsAG2 and LsDEF1 contain 19 and 25 phosphorylation sites, and both contain a conserved MADS-MEF2-like domain and a K-box domain, but there are some differences between the two protein sequences, which we observed in the start and end positions of the two domains (Table S3, Figure S1). The secondary structures of LsAG2 and LsDEF1 were predicted, demonstrating that they both have four types of coils: α-helix, β-sheet, extended chain, and random coil, of which the main components are α-helix (46.41%–51.98%) and random coil (29.11%–31.75%) (Table S4). A phylogenetic tree was constructed by comparing the homologous proteins of LsAG2 and LsDEF1 with other homologous protein sequences, and we found that both LsAG2 and LsDEF1 were closest to the homologous protein in pomegranate in terms of genetic distance (Figure 1). That is, LsAG2 and LsDEF1 have the typical characteristics of MADS-box family transcription factors, and both belong to the MADS-box family.

*3.2. Spatiotemporal Specificity Expression Analysis of LsAG2 and LsDEF1*

In order to study the spatiotemporal specificity expression of *LsAG2* and *LsDEF1*, the expression levels of *LsAG2* and *LsDEF1* in different flower development stages and different floral organ tissues of the SPL and DPL were determined using RT-qPCR. The results showed that *LsAG2* was not expressed in the calyx of SPL, and the expression level was extremely low in the single-petaled pistils and petals; however, its expression was significantly up-regulated in the pistils and petals of DPL. In the S3 stage, the expression level of *LsAG2* in the SPL was significantly higher than in the other three periods, while in the DPL, the expression level of *LsAG2* in the S4 stage was slightly higher than in the other three stages (Figure 2a,b; Table S5). *LsDEF1* was very low or not expressed in the calyces and pistils of the DPL and SPL and was relatively high in the petals. The expression level of *LsDEF1* in the stamens of the DPL was higher than that in the stamens of SPL but slightly lower than that in the petals of SPL. *LsDEF1* was expressed in each stage of flower development in the DPL and showed a gradual upward trend, reaching the highest value in the S4 stage. In the SPL, the expression level of *LsDEF1* first increased and then decreased (Figure 2c,d; Table S5).

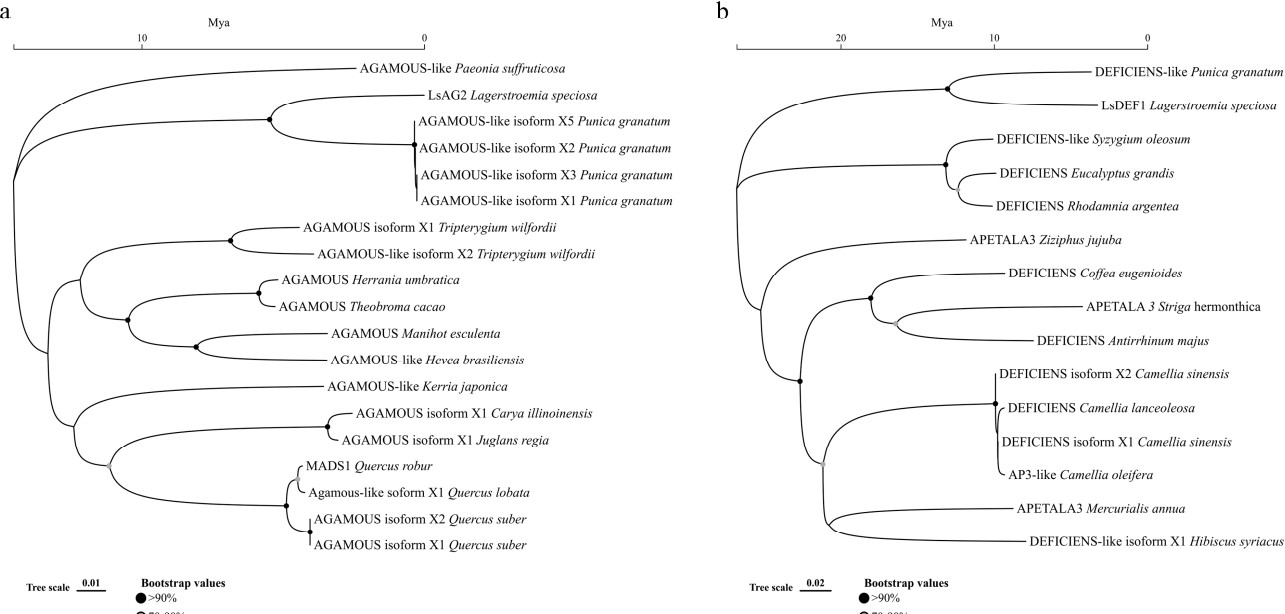

**Figure 1.** Phylogenetic analysis of LsAG2, LsDEF1, and other MADS-box proteins from different species. (**a**) Phylogenetic analysis of LsAG2 and MADS-box proteins of *Paeonia suffruticosa*, *Punica granatum*, *Tripterygium wilfordii*, *Herrania umbratical*, *Theobroma cacao*, *Manihot esculenta*, *Hevea brasiliensis*, *Kerria japonica*, *Carya illinoinensis*, *Juglans regia*, *Quercus robur*, *Quercus lobata*, and *Quercus suber*. (**b**) Phylogenetic analysis of LsDEF1 and MADS-box proteins of *Punica granatum*, *Syzygium oleosum*, *Eucalyptus grandis*, *Rhodamnia argentea*, *Ziziphus jujuba*, *Coffea eugenioides*, *Striga hermonthica*, *Antirrhinum majus*, *Camellia lanceoleosa*, *Camellia sinensis*, *Camellia oleifera*, *Mercurialis annua*, and *Hibiscus syriacus*.

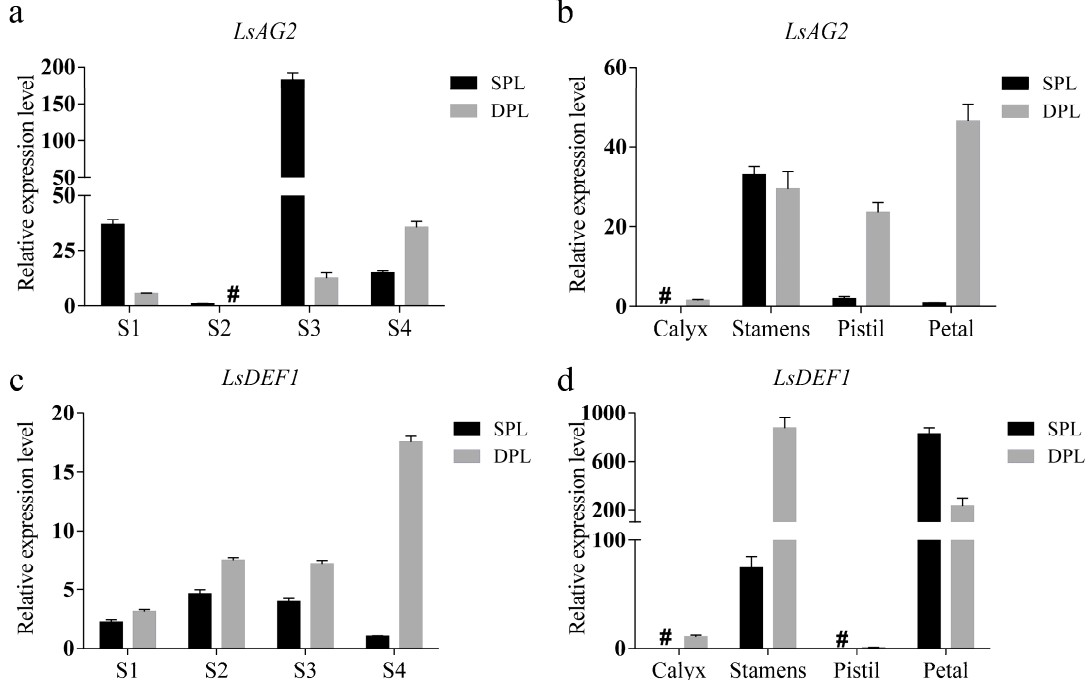

**Figure 2.** The expression patterns of *LsAG2* and *LsDEF1*. (**a**) Expression level of *LsAG2* during different developmental stages. (**b**) Expression level of *LsAG2* in different flower tissues. (**c**) Expression level of *LsDEF1* during different developmental stages. (**d**) Expression level of *LsDEF1* in different flower tissues. S1 to S4, four different development stages of flower budding. SPL, single-petaled *L. speciosa*;

DPL, double-petaled *L. speciosa*. In the same type of tissue or developmental stage, the sample with the lowest expression was selected as the control, and the gene expression levels were calculated using the $2^{-\Delta\Delta Ct}$ method. The symbol '#' indicates that the gene cannot be detected.

### 3.3. Subcellular Localization of LsAG2 and LsDEF1

The *LsAG2-GFP* and *LsDEF1-GFP* fusion genes (pSuper-*35S::LsAG2-GFP*, pSuper-*35S::LsDEF1-GFP*) were used to examine the subcellular localization of LsAG2 and LsDEF1 transiently expressed in *N. benthamiana* leaves. The results showed that the GFP signals of LsAG2-GFP and LsDEF1-GFP were detected only in the nucleus of the tobacco cells, while the GFP signal of the empty plasmid control was distributed throughout the whole cell (Figure 3).

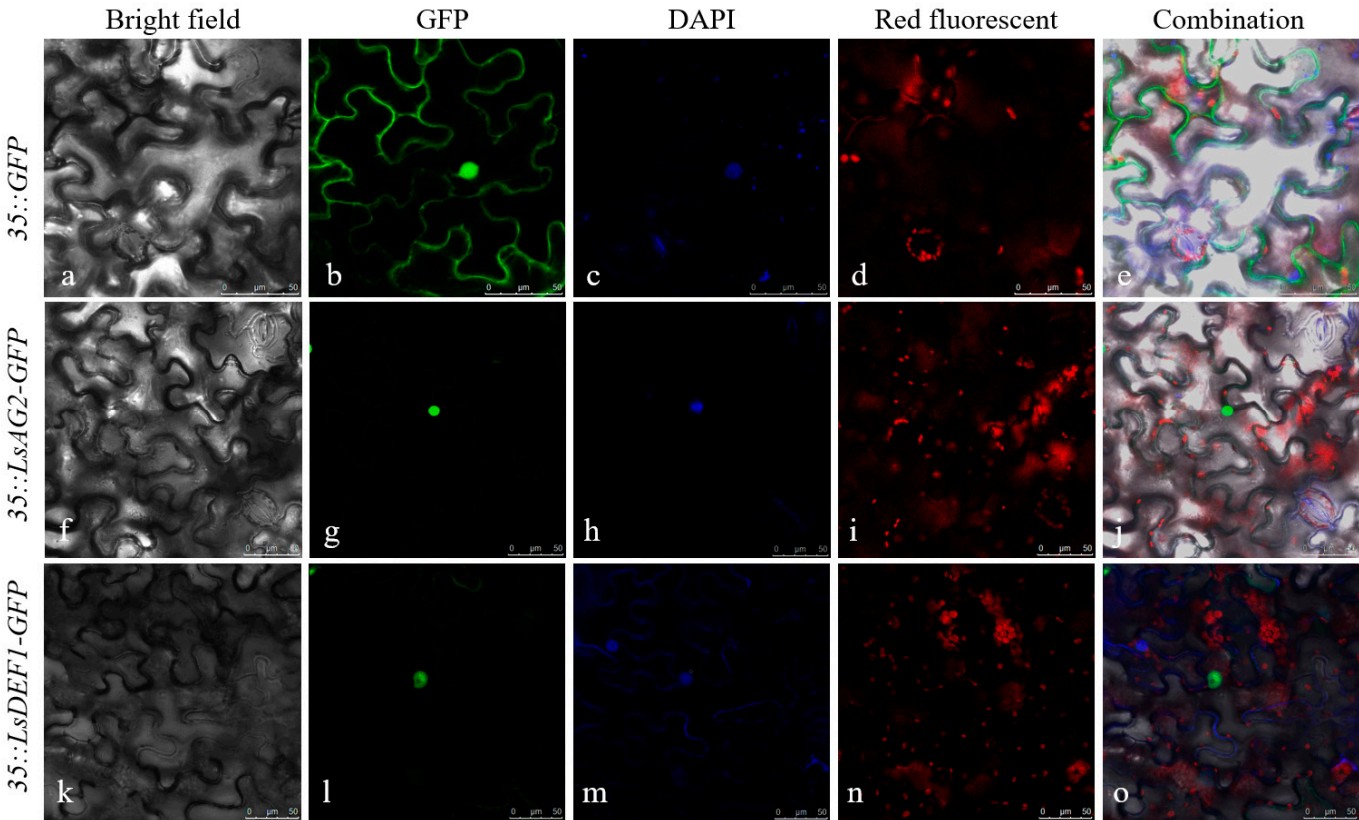

**Figure 3.** Subcellular localization of LsAG2 and LsDEF1 proteins in tobacco leaf cells. (**a**–**e**) Subcellular localization of GFP. (**f**–**j**) Subcellular localization of LsAG2-GFP. (**k**–**o**) Subcellular localization of LsDEF1-GFP. Bright field: white light. GFP: green fluorescence signal. DAPI: 4′,6-diamidino-2-phenylindole (DAPI) nuclear DNA fluorescent stain. Red fluorescent: chloroplast auto-fluorescence. Combination: combined signals of different fluorescence.

### 3.4. Phenotype of Overexpression of LsAG2 and LsDEF1 in Arabidopsis

*LsAG2* and *LsDEF1* were transferred into *A. thaliana* using *A. tumefaciens*-mediated floral organ dipping. Finally, 15 and 16 transgenic lines overexpressing the *LsAG2* and *LsDEF1* genes were obtained (Figure S2), respectively. Under long-day light conditions, the wild-type *A. thaliana* took approximately 45 d from sowing to flowering. However, many *35S::LsAG2* and *35S::LsDEF1* T$_2$ transgenic lines emerged in the flowering period, approximately 30 d after sowing (Table S6), and very few lines appeared before the seedlings were transplanted (just about 10 d). Compared to the wild-type, some *35S::LsDEF1* T$_2$ transgenic lines had more flowers and a longer flowering time.

In addition to the significant changes in the flowering period, the leaves and flowers of the transgenic lines also changed. The *35S::LsAG2* transgenic plant had smaller flower

diameters than the wild-type *A. thaliana* (Figure 4a,e,f), and similar abnormal flower phe-
notypes, such as smaller petals, a sepal-like shape, hypoplasia, or the absence of petals,
enlarged stamens, and less loose powder, were detected in the *35S::LsAG2* transgenic plants
(Figure 4b–d). These transgenic plants were still fertile. On the other hand, abnormal
development of the flower organs may be the reason of for *35S::LsDEF1*'s infertility, as a
few of the plants lacked pistils, and the number of stamens was 7–10 (Figure 4j,k). A few
were missing petals or showed malformed development of the entire flower (Figure 4l,m),
and in some lines, the number of pistils increased to four, and the number of stamens in-
creased to eight. Both the *35S::LsAG2* and *35S::LsDEF1* transgenic plants showed significant
early flowering (Figure 4g,h,n), and the $T_2$ generation of the *35S::LsDEF1* transgenic plants
showed a lack of fruiting, with a larger flower volume and longer flowering period than
the wild-type *A. thaliana* (Figure 4i).

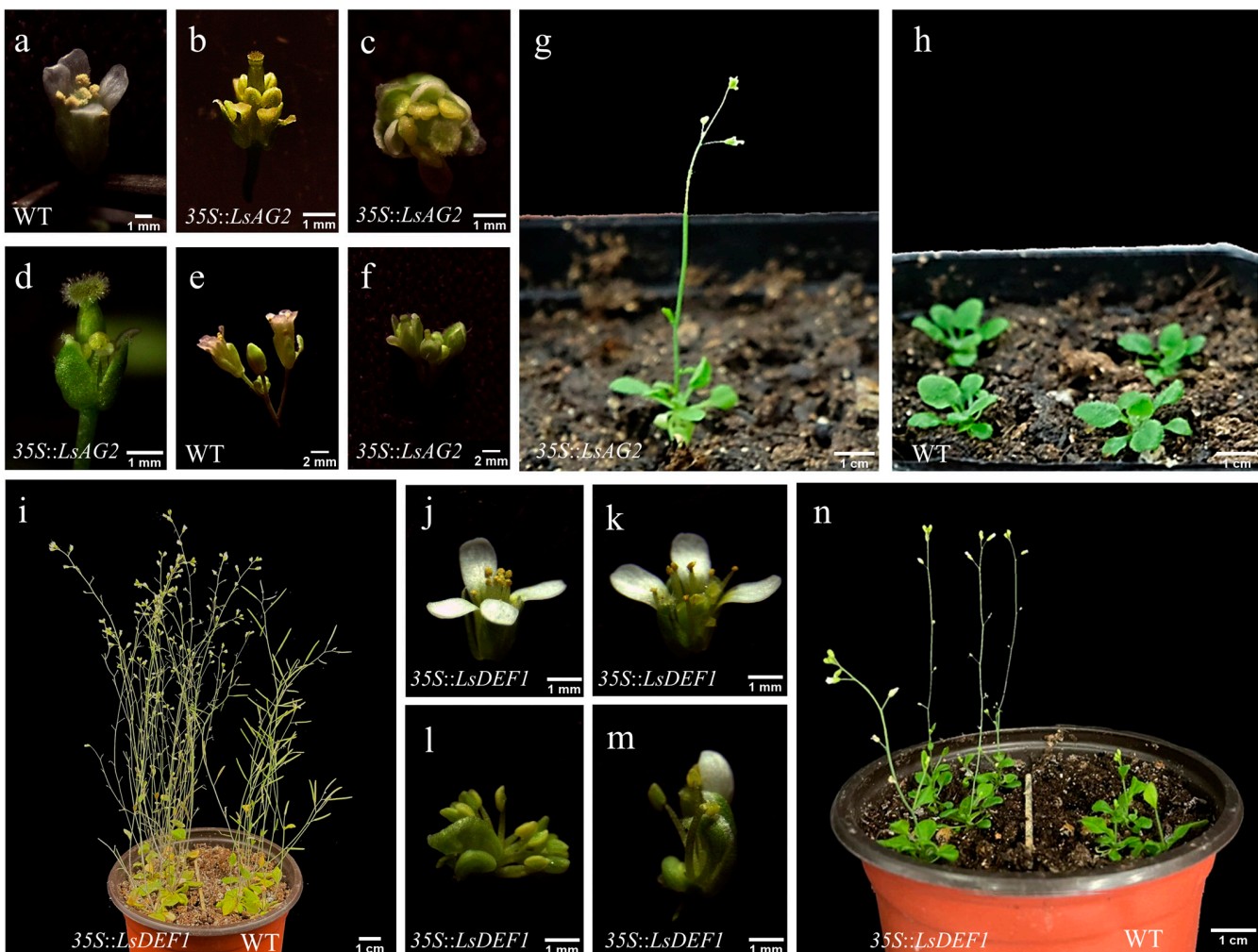

**Figure 4.** Phenotypes of overexpression of *LsAG2* and *LsDEF1* in *Arabidopsis*. (**a**) Flower of wild-
type *A. thaliana*. (**b**–**d**) Flowers of *35S::LsAG2* transgenic *A. thaliana*. (**e**) Inflorescence of wild-type
*Arabidopsis*. (**f**) Inflorescence of *35S::LsAG2* transgenic *A. thaliana*. (**g**) A 30-day-old *35S::LsAG2*
transgenic *A. thaliana* starts flowering. (**h**) A 30-day-old wild-type *A. thaliana*. (**i**) The *35S::LsDEF1*
transgenic *A. thaliana* has a longer flowering time and more flowers than the wild-type. (**j**–**m**) Flowers
of *35S::LsDEF1* transgenic *A. thaliana*. (**n**) A 35-day-old *35S::LsDEF1* transgenic *A. thaliana* starts
flowering earlier than the wild-type.

### 3.5. Detection of Endogenous Genes Related to the Flower Time and Flower Development

In order to identify whether the transcription levels of endogenous genes related to
the flower time and flower development in *A. thaliana* are affected by the overexpression of

the *LsAG2* and *LsDEF1*, RT-qPCR was used to detect the transcription levels of endogenous genes (*AtAP1*, *AtLFY*, *AtFLC*, *AtFT*, *AtPI*, and *AtFUL)* in transgenic *A. thaliana*. The results showed that in the overexpression of the *LsAG2* gene in *A. thaliana*, the expression levels of endogenous genes significantly decreased, except for *AtPI* and *AtFUL*. Upon overexpressing *LsDEF1* in *A. thaliana* lines, the expression level of *AtLFY* was slightly down-regulated, while the expression levels of *AtAP1*, *AtPI*, *AtFLC*, and *AtFUL* were up-regulated (Figure 5; Table S7).

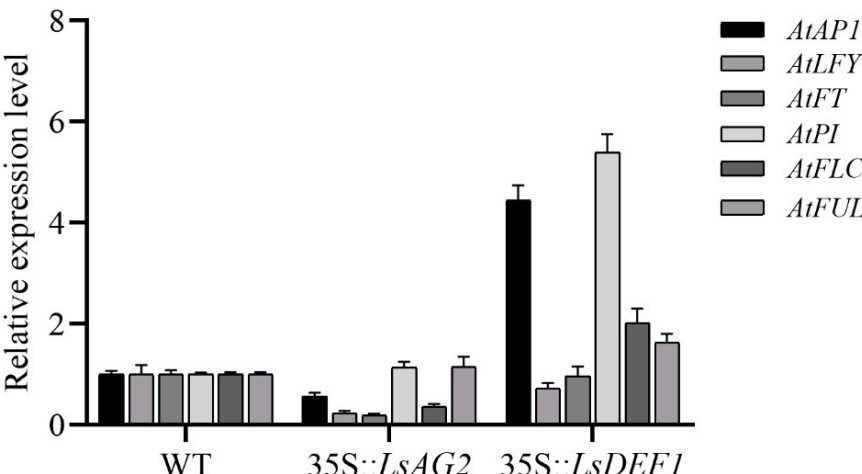

**Figure 5.** Expression levels of endogenous genes related to the flower time and flower development. The wild-type was selected as the control, and the gene expression levels were calculated using the $2^{-\Delta\Delta Ct}$ method.

## 4. Discussion

*Lagerstroemia speciosa* is an ornamental tree in summer, originating from Asia. The analysis of its flower development process is of great significance for the flowering period and flower type in breeding. Using the DPL and SPL as materials, the CDS, full-length DNA, and the promoter sequence approximately 1800 bp upstream of the *LsAG2* and *LsDEF1* genes were cloned by PCR. Previous studies have found that the formation of double-petal phenomena in many plants is closely related to structural differences in MADS-box or AP2 family genes. For example, the formation of the double rose is related to the deletion of the target site of *miR172* on *AP2* [24]. In sakura (*Prunus lannesiana*), there is a 190 bp skip reading in the exon of *AGAMOUS* (*PrseAG*), and the protein encoded by it also shows several amino acid differences from the protein encoded by the same gene in single-petaled *P. lannesiana* [25]. Comparing the CDS and DNA sequences cloned from the DPL and SPL, we found that there was no difference between the sequences, indicating that the formation of the stamen petalization of *L. speciosa* may not be caused by the differences in the gene structures of *LsAG2* and *LsDEF1*. Meanwhile, there was no difference in the upstream promoter region sequence, indicating that the differences in the gene expression levels of *LsAG2* and *LsDEF1* may be caused by other unknown regulation-related genes.

The *LsAG2* is the homologous gene of C-type gene *AG* in *A. thaliana*. Its expression pattern in different flower organs of the SPL showed more resemblance to the characteristics of the C-type gene in the ABCDE model, and the expression pattern in different floral organs of the DPL was very different from that in the SPL, indicating that the ectopic expression of *LsAG2* may be closely related to the formation of the DPL. *LsDEF1* is a B-type gene in the flower development model, and its expression pattern in different floral organs of *L. speciosa* was completely consistent with the characteristics of B-type genes. The *DEFICIENS* (*DEF*) homologous genes in other species also show the same trend [26], suggesting that the function of *DEF* may be conserved in angiosperms. In addition, the results of subcellular localization showed that the fusion proteins formed by the LsAG2-GFP and LsDEF1-GFP proteins were all localized in the nucleus, indicating that the proteins should have the

function of transcription factors and may participate in the related processes of *L. speciosa* flower development.

*AG* has important and conserved functions in floral organ development in a variety of plants and is closely related to the double petal formation of angiosperm. For example, the overexpression of *JcAG* from Jatropha (*Jatropha curcas*) in *A. thaliana* also causes early flowering [27]. The overexpression of *TeAG1* and *TeAGL11-1* of marigold (*Tagetes erecta*) leads to leaf curling and early flowering in *A. thaliana* [28]. Here, the *35S::LsAG2* transgenic plants exhibited multiple phenotypes, such as flower aberration (shortened petal length, sepalization, or even its absence), early flowering, leaf curling, dwarf plants, and a reduced seed yield, indicating that *LsAG2* is widely involved in a variety of lifecycle activities, including flower development, in *A. thaliana*. The overexpression the *LsDEF1* in *A. thaliana* resulted in the phenotypes of degenerated pistils and increased stamens in the flowers. The overexpression of *BcAP3* (*APTEROUS3*) of turnip (*Brassica rapa* ssp.) in *A. thaliana* can lead to abnormal anther development and low pollen viability, resulting in male sterility [29]. The ectopic expression of *FaesAP3* of buckwheat (*Fagopyrum esculentum*) can rescue the stamen development and lack of petal development in *A. thaliana ap3* mutants [30], indicating that *DEF1/AP3* may be conserved in different plants.

In order to explore the reason for the double petalization of DPL, we measured the expression levels of *LsAG2* and *LsDEF1* in different tissues and performed functional verification on *A. thaliana*. RT-qPCR showed that the expression level of *LsAG2* in the petals of the DPL was significantly higher than that in the petals of the SPL, and the main phenotype of *35S::LsAG2* transgenic *Arabidopsis* is the distortion of petals. The expression level of *LsDEF1* in the stamens of the DPL was significantly higher than that in the stamens of the SPL, and the main phenotype of *35S::LsDEF1* transgenic *Arabidopsis* is the increase in the number of stamens, indicating that these two genes are, indeed, involved in the process of petal morphological transformation, but there is still a lack of key steps that result in petaloid stamens or increase the number of petals. A large number of studies have shown that changes in the expression patterns of *AG* homologous genes can lead to the formation of double flowers. This change in expression patterns includes the loss of gene expression [31,32] or the loss of key regions of the protein during transcription [25,33]. Therefore, future research should focus on exploring the effects of changes in the expression patterns of *AG* homologous genes on *L. speciosa* flower development and the screening of interacting proteins using yeast two-hybrid technology to analyze the mechanism of double flower formation in *L. speciosa*.

**5. Conclusions**

The *LsAG2* and *LsDEF1* genes of *L. speciosa* were cloned, and their expression patterns were verified by RT-qPCR. The results of subcellular localization showed that both genes were localized in the nucleus, and when *LsAG2* and *LsDEF1* were overexpressed in *A. thaliana*, the floral organs of the transgenic plants were changed, and the expression levels of endogenous genes related to the flower time and flower development also changed. In brief, *LsAG2* and *LsDEF1* have important roles in *L. speciosa* flower development, and the results of this research provide a theoretical basis for the regulation of flowering time and flower development in *L. speciosa*.

**Supplementary Materials:** The following supporting information can be downloaded at: https://www.mdpi.com/article/10.3390/agronomy13040976/s1, Figure S1: Bioinformatics analysis of *LsAG2* and *LsDEF1*; Figure S2: PCR detection of transgenic *A. thaliana*; Table S1: The primer sequences for cloning, RT-qPCR and vector construction; Table S2: Analysis of promoter cis-regulatory elements; Table S3: Physicochemical properties of MADS-box proteins of *L. speciosa*; Table S4: Secondary structures of MADS-box proteins of *L. speciosa*; Table S5: Relative expression data of *LsAG2* and *LsDEF1* in different samples; Table S6: Flowering time of *LsAG2* and *LsDEF1* transgenic *A. thaliana* plants under long-day conditions; Table S7: Relative expression data of *LsAG2* and *LsDEF1* in wild-type and transgenic *Arabidopsis*.

**Author Contributions:** Funding acquisition, T.Z. and Q.Z.; investigation, Z.L.; methodology, L.Y. and Z.L.; project administration, Z.L., T.Z. and Q.Z.; resources, T.Z., J.W. and T.C.; supervision, T.Z., J.W. and T.C.; validation, L.Y.; visualization, L.Y. and Z.L.; writing—original draft, L.Y. and Z.L.; writing—review and editing, T.Z. and Q.Z. All authors have read and agreed to the published version of the manuscript.

**Funding:** The research was supported by the Beijing Municipal Science and Technology Project (grant No. Z181100002418006), Beijing High-Precision Discipline Project, Discipline of Ecological Environment of Urban and Rural Human Settlements and Special Fund for Beijing Common Construction Project.

**Data Availability Statement:** Not applicable.

**Conflicts of Interest:** The authors declare no conflict of interest.

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
