# Peer review of "Overexpression of Two MADS-Box Genes from Lagerstroemia speciosa Causes Early Flowering and Affects Floral Organ Development in Arabidopsis"

_agronomy, doi:10.3390/agronomy13040976_

Round 1

Reviewer 1 Report

Reviewer's comments:

Authors of this manuscript have studied ABCDE members of the MADS-box family in floral organ development in L. speciosa. The authors analyzed the expression of two MADS-box genes (LsAG2 and LsDEF1) that were isolated from L. speciosa. In addition, the authors studied the phenotypes of overexpressed lines of those two genes in Arabidopsis.

Although this is an interesting topic, I have major concerns with respect to compilation of the manuscript, language editing, and some missing information. Please go through PDF files where I inserted my comments and resubmit for further review. 

Please have the manuscript edited with the help of native English speaking scientific editor. 

Author Response

Dear reviewer,

Thank you very much for your letter and patient work for our manuscript. Those suggestions are all valuable and very helpful for revising and improving our paper. We have studied comments carefully and have made corrections that we hope to meet with approval. The manuscript was checked and edited for proper English language, grammar, punctuation, spelling, and overall style by MDPI editing services (english-61787). The main revises and the responses to comments are as follows:

Reviewer 1#

Authors of this manuscript have studied ABCDE members of the MADS-box family in floral organ development in L. speciosa. The authors analyzed the expression of two MADS-box genes (LsAG2 and LsDEF1) that were isolated from L. speciosa. In addition, the authors studied the phenotypes of overexpressed lines of those two genes in Arabidopsis.

Although this is an interesting topic, I have major concerns with respect to compilation of the manuscript, language editing, and some missing information. Please go through PDF files where I inserted my comments and resubmit for further review.

Please have the manuscript edited with the help of native English speaking scientific editor.

Response: The manuscript was checked and edited for proper English language, grammar, punctuation, spelling, and overall style by MDPI editing services (english-61787).

In PDF File comments:

1.L85 What does Powdered mean?

Response: It was written incorrectly, which should be “powered”. We had revised it in the text.

2.L90 This title should go after the RNA extraction, isn't it?

Response: Thank you. We have adjusted the position of the two sections in the text.

3.L02,L107 were-was

Response: We had revised it in the text.

4.L110-114 poor English. Please edit with native speaking editorial assistance.

Response: Thank you. The manuscript was checked and edited for proper English language, grammar, punctuation, spelling, and overall style by MDPI editing services (english-61787).

5.L115 What is the expansion? It has to be written as RT-qPCR.

Response: We have changed all “qRT-PCR” to “RT-qPCR” in the text.

6.L117 include the country name too as you did for other products.

Response: The country name of reagents and instruments were added in the text.

7.L128 it must be "synthesized" not designed.

Response: This sentence is improved with the “synthesized”.

  1. L128 2^–∆∆Ct

Response: We have changed all “delt delt Ct” to “2^–∆∆Ct” in the text.

9.L138 L139 was-were

Response: We had revised the word format error in the text.

10.L141 Is it a new method?

Response: Thank you. This is an improved method. We have re-organized the language and improved the content of this sentence.

11.L152 Did you transform the vectors directly into Arabidopsis plants?

Response: Thank you very much. What we wrote is wrong. We have reorganized the sentence in the text.

12.L175-177 This sentence should have respectively. Grammatical error.

Response: Thank you. We have reorganized the sentence in the text.

13.L193 italics

Response: We have changed it to italic.

14.L226 What did you use as a control for the nucleus?

Response: We have used DAPI to dye the nucleus in pre-experiment stage, but the fluorescence signal of the nucleus of the successfully transformed cells will become weak, and can't get very clear and beautiful pictures.

15.L232 What is the use of Red Fluorescent here?

Response: Red Fluorescent is the spontaneous fluorescence of chloroplast at 488 nm.

16.L225 Label the panel with gene notation. That will be easy to follow than referring the legend everytime.

Response: Thank you. I have added gene notations below each small figures.

17.L225 Please pay attention to the scientific names of organisms, gene names in italics.

Response: Thank you. I have modified the species and gene name format in italic.

18.Check all references.

Response: All references have been carefully checked and re-edited.

19.Please arrange the figure legends close to the figure. Thanks.

Response: We had the arrange the figure legends under the figure in one page.

  1. FigS2, Where is negative control?

Response: The negative control was the WT (wild type) in the last line in FigS2a.

Reviewer 2 Report

Comments for authors, kindly find at the attachment.

Author Response

Dear and reviewer,

Thank you very much for your letter and patient work for our manuscript. Those suggestions are all valuable and very helpful for revising and improving our paper. We have studied comments carefully and have made corrections that we hope to meet with approval. The manuscript was checked and edited for proper English language, grammar, punctuation, spelling, and overall style by MDPI editing services (english-61787). The main revises and the responses to comments are as follows:

Reviewer 2#

Comments for authors, kindly find at the attachment.

1.L88 Flower bud of different development stages (S1: buds 4 - 6 mm in diameter, S2: buds 6-10 mm in diameter, S3: buds 10-14 mm in diameter, S4: full bloom but not powdered) and different floral organ tissues (calyx, petals, stamens / petalized stamens, pistils) at S4 stage of the SPL and DPL were collected as our previous study (Hu et al., 2019)

Please correct the type of citation.

Response: I have corrected the citation format in [20].

2.L116-117 Total RNA was extracted from 16 samples of flower bud of different development stages and different floral organ tissues using a RNAprep Pure Plant Kit.

Please specify, there were DPL or SPL, which sample and which stages? 16 samples from 4 stages means 4 samples for each stage, but I guess.

Response: Thank you very much. We have added descriptions of these 16 samples to the text. 16 samples were from 4 different developmental stages (S1 to S4) and 4 different floral organ tissues (calyx, stamen, pistil, petal) of the DPL and SPL.

3.L127-128  Gene expression levels were calculated using the 2-delta-delta Ct method.

Please add the description, how the gene expression levels were calculated, e.i. which samples were control samples.

Response: Thank you very much. We use 2 −∆∆Ct method to calculated the gene expression levels, EF-1α gene of L. speciosa was employed as a reference gene, and in the same type of tissue, the sample with the lowest expression was selected as the control. We have added this sentence in the text.

4.L138-141 The recombinant plasmid pSuper1300-LsAG2-GFP, pSuper1300-LsDEF1-GFP was transformed into Agrobacterium tumefaciens competent cells GV3101, recombinant plasmid and empty vector plasmid were transient expressed in healthy Nicotiana benthamiana leaves by injection infection

Agrobacterium tumefaciens – please correct for italic

Response: We had correct the “Agrobacterium tumefaciens” in italic.

  1. Line 193: (a) Phylogenetic analysis of LsAG2 and MADS-box proteins of Paeonia suffruticosa

Please correct Paeonia suffruticosa for italic.

Response: We had correct the “Paeonia suffruticosa” in italic.

6.L216-219  Figure 2 The expression patterns of LsAG2 and LsDEF1. (a) Expression level of LsAG2 during different developing stages; (b) Expression level of LsAG2 in different flower tissues; (c) Expression level of LsDEF1 during different developing stages; (d) Expression level of LsDEF1 in different flower tissues. S1 to S4, four different development stages of flower bud; SPL, the single- petaled L. speciosa; DPL, double- petaled L. speciosa.

Please add in the description, how relative expression level was calculated. Description should be self-explanatory.

Response: Thank you very much. In the same type of tissue, the sample with the lowest expression was selected as the control. We have added this sentence in the text.

7.L318-325

 qRT-PCR showed that the expression level of LsAG2 on petals in the DPL was significantly higher than that in the SPL, and the expression level of LsDEF1 on stamens in the DPL was significantly higher than that in the SPL, and when LsAG2 and LsDEF1 were overexpressed in A. thaliana, The main phenotype of 35S::LsAG2 transgenic Arabidopsis is the distortion of petals, and the main phenotype of 35S::LsDEF1 transgenic Arabidopsis is the increase in the number of stamens, indicating that these two genes are indeed involved in the process of petal morphological transformation, but there is still a lack of key steps that make stamen petaloid or increase the number of petals.

I have some doubts about the correlation between the changes in the expression level, the phenotype changes and the role of these genes. Could you describe this relation more clearly?

Response: Thank you. I have stated trait correlations more clearly in the text. The expression level of LsAG2 in the petals of DPL was higher than that of SPL, and its transgenic plants also showed changes in the floral part, and the expression level of LsDEF1 was high in stamens, and the transgenic plants also showed corresponding changes. We have reorganized these sentences in the text.

8.L272 Figure 5. Expression level of endogenous genes related to flower time and flower development.

Please add in the description, how relative expression level was calculated. Description should be self-explanatory.

Response: Thank you. Thank you very much. In the same type of tissue, the sample with the lowest expression was selected as the control. We have added this sentence in the text.

Figure S2: PCR detection of transgenic A. thaliana

Please add the description what there are the 1-15 samples in the “a” picture and 1-16 samples in the “b” picture. Why do the PCR products differ in the length on the gel (picture “a”)?

Response: Thank you very much. We have added “a” “b” in the Figure S2. Since we use Gelred loading Dye for gel electrophoresis, the DNA Marker in Figure S2 a is slightly distorted, and all the tested samples are also slightly distorted, so some PCR products differ in the length.

 Table S1: The primer sequences for cloning, qRT-PCR and vector construction,

Please add the annealing temperature to each primer pairs.

Response: Thank you very much. I have added the Annealing temperature to the Table S1.

Round 2

Reviewer 1 Report

1. In the references, the authors should pay attention for the italics notations. There are still few errors: one example  #11. cis motifs

2. What is the positive control for the nuclear localization in the subcellular localization experiment? Did the authors respond to this question somewhere in the manuscript, especially in the materials and methods and also in the figure legend? GFP alone localized throughout the cell. However, how did the authors confirm that AG2 and DEF1 were detected only in the nucleus without positive control? Please justify. Thanks.

Author Response

Dear reviewers,

Thank you very much for your letter and patient work for our manuscript. Those suggestions are all valuable and very helpful for revising and improving our paper. We have studied comments carefully and have made corrections that we hope to meet with approval. The manuscript was checked and edited for proper English language, grammar, punctuation, spelling, and overall style by MDPI editing services (english-61787). The main revises and the responses to comments are as follows:

Reviewer 1#

  1. In the references, the authors should pay attention for the italics notations. There are still few errors: one example #11. cis motifs

Response: Thank you. All references have been rechecked and corrections made where italics were required.

  1. What is the positive control for the nuclear localization in the subcellular localization experiment? Did the authors respond to this question somewhere in the manuscript, especially in the materials and methods and also in the figure legend? GFP alone localized throughout the cell. However, how did the authors confirm that AG2 and DEF1 were detected only in the nucleus without positive control? Please justify. Thanks.

Response: Thank you very much. We have added a statement about “4',6-diamidino-2-phenylindole (DAPI) was used to dye the nuclear DNA as a positive control for nuclear localization” in the Materials and methods section, and we have changed the Figure3 with DAPI staining in the Result section.

Reviewer 2 Report

Kindly find the comments in the attachment.

Author Response

Dear reviewer,

Thank you very much for your letter and patient work for our manuscript. Those suggestions are all valuable and very helpful for revising and improving our paper. We have studied comments carefully and have made corrections that we hope to meet with approval. The manuscript was checked and edited for proper English language, grammar, punctuation, spelling, and overall style by MDPI editing services (english-61787). The main revises and the responses to comments are as follows:

Reviewer 2#

Kindly find the comments in the attachment.

Lines 165-166: In the same type of tissue, the sample with the lowest expression was selected as the control.

If I good understand, the control sample for LsAG2 is SPL Calyx and for LsDEF1 is SPL or DPL pistil?

Response: Thank you very much. We had redrawn the Figure 2. Each small figure is independent, and the lowest expression is used as a reference. We have revised the sentence in the text.

Lines 259-261: The results showed that LsAG2 wais not expressed in the calyx in theof SPL, and the expression level wais extremely low in the single-petaled pistils and petals;, however, its expression was significantly up-regulated in the pistils and petals of DPL. In the S3 stage, the expression level of LsAG2 in the SPL was significantly higher than in the other three periods, while in the DPL, the expression level of LsAG2 in the S4 stage was slightly higher than in other three stages (Fig. 2a, b).

Please add to the Materials and methods how the statistics on the gene expression data was done. Please also add in the Supplementary material the CT data or other gene expression data for each samples, for each tissue, for each stage, for each gene.

Response: Thank you very much. We have added the calculation procedure for 2 −∆∆Ct method in the Materials and methods section. And we add the gene expression data for each samples, for each tissue, for each stage, for each gene in Table S5 and Table S6. We also revised the sentence in the text.

Line 311: Figure 5. The sample with the lowest expression was selected as the control.

If I good understand, the control samples for AtAP1, AtLFY, AtFT, AtFLC are the same genes for 35S::LsAG2?

Response: Thank you. We made mistakes in writing. We used the gene expression levels of wild-type Arabidopsis as a control, our purpose is to explore the changes in the expression levels of these genes in transgenic Arabidopsis. We have added this sentence in the text.

Figure S2: PCR detection of transgenic A. thaliana

Still no description what there are the 1-15 samples in the “a” picture and 1-16 samples in the “b” picture.

Response: Thank you. We had added and revised the legend of the Figure S2.

Table S1: The primer sequences for cloning, qRT-PCR and vector construction,

Still no the annealing temperature to each primer pairs.

Response: Thank you. I have added the Annealing temperature to the Table S1 as following:

Table S1: The primer sequences for cloning, RT-qPCR and vector construction

Primer name

Primer sequence (5’- 3’)

Annealing temperature

Purpose

LsAG2-F

ATGATGTTCCCGAGCCAATCGAG

58 °C

gene cloning

LsAG2-R

CTAAACTAACTGGAGGGCCATCTG

LsDEF1-F

ATGACGAGAGGGAAGATTCAGATCAA

58 °C

LsDEF1-R

TCACTGGTCGAGCAAGGGATAGGT

EF-1α F

GACTGTGCTGTGCTCATC

57 °C

qRT-PCR

EF-1α R

GTGGCATCCATCTTGTTG

qLsAG2-F

TTGCGGAGAGTGAGAGAA

58 °C

qLsAG2-R

TGAGTTGGAGGCTGAGAA

qLsDEF1-F

GTCGTGGTGGTGGGGCTGATGTA

60°C

qLsDEF1-R

AACCGGCAGGTCACCTACTCGAAG

1304LsAG2-F

GAACACGGGGGACTCTTGACATGATGTTCCCGAGCCAATC

60°C

cloning fragment

1304LsAG2-R

ATTCGAGCTGGTCACCTAAACTAACTGGAGGGCCATCTGG

1304LsDEF1-F

GAACACGGGGGACTCTTGACATGACGAGAGGGAAGATTCA

60°C

1304LsDEF1-R

ATTCGAGCTGGTCACTCACTGGTCGAGCAAGGGATAGGTC

AtActin-F

GGAGCTGAGAGATTCCGTTG

58 °C

Arabidopsis qRT-PCR

AtActin-R

GGTGCAACCACCTTGATCTT

AtFUL-F

ATCTCTGTTCTCTGCGATGCT

59 °C

AtFUL-R

AGCGTTCAAGTATCCTCTCCAT

AtAP1-F

CTCTGTTCTCTGTGATGCTGAA

58 °C

AtAP1-R

AGCGTTCAAGTATCTTCTCCAT

AtLFY-F

GGATAACGGCAACGGAGGTA

60°C

AtLFY-R

AAGAAGGAACTCACGGCATTG

AtFT-F

AGAGGTGACTAATGGCTTGGA

59 °C

AtFT-R

TTGCTAGGACTTGGAACATCTG

AtPI-F

AAGTCCGAGACCACCAGATG

59 °C

AtPI-R

CCTCTTGCGTTGCTTGCTATA

AtFLC-F

AGCCAAGAAGACCGAACTCAT

59 °C

AtFLC-R

AGCAGGTGACATCTCCATCTC

Round 3

Reviewer 2 Report

Kindly find the comments in the attachment.

Author Response

Dear reviewer,

Thank you very much for your letter and patient work for our manuscript. Those suggestions are all valuable and very helpful for revising and improving our paper. The main revises and the responses to comments are as follows:

Reviewer 2#

1.Lines 240-243: The results showed that LsAG2 was not expressed in the calyx in of SPL, and the expression level was extremely low in the single-petaled pistils and petals;, however, its expression was significantly up-regulated in the pistils and petals of DPL. In the S3 stage, the expression level of LsAG2 in the SPL was significantly higher than in the other three periods, while in the DPL, the expression level of LsAG2 in the S4 stage was slightly higher than in other three stages (Fig. 2a, b).

Please add to the Materials and methods how the statistics on the gene expression data was done.

Response: Thank you. The statistical method of gene expression data has been added to the MM section. As following: “Statistical analyses were performed using the SPSS 19.0 (SPSS, Chicago, Illinois, USA). ANOVA was used for multiple-group comparisons, and statistical significance (p < 0.05) was determined by Student’s test.”

2.Table S5: Relative expression data of LsAG2 and LsDEF1 in different samples

Table S6: Relative expression data of LsAG2 and LsDEF1 in wild type and transgenic Arabidopsis

Please add the description, how is the unit of the relative expression.

Response: Thank you. We had added the unit (ΔΔCt) of the relative expression in the Table S5 and S6.
